# Enhanced Stability of Li-RHC Embedded in an Adaptive TPX™ Polymer Scaffold

**DOI:** 10.3390/ma13040991

**Published:** 2020-02-22

**Authors:** Thi Thu Le, Claudio Pistidda, Clarissa Abetz, Prokopios Georgopanos, Sebastiano Garroni, Giovanni Capurso, Chiara Milanese, Julián Puszkiel, Martin Dornheim, Volker Abetz, Thomas Klassen

**Affiliations:** 1Institute of Materials Research, Materials Technology, Helmholtz-Zentrum Geesthacht GmbH, Max-Planck-Strasse 1, D-21502 Geesthacht, Schleswig-Holstein, Germany; giovanni.capurso@hzg.de (G.C.); julianpuszkiel1979@gmail.com (J.P.); martin.dornheim@hzg.de (M.D.); thomas.klassen@hzg.de (T.K.); 2Institute of Polymer Research, Helmholtz-Zentrum Geesthacht, Max-Planck-Straße 1, 21502 Geesthacht, Schleswig-Holstein, Germany; clarissa.abetz@hzg.de (C.A.); prokopios.georgopanos@hzg.de (P.G.); volker.abetz@hzg.de (V.A.); 3Department of Chemistry and Pharmacy and INSTM, University of Sassari, Via Vienna 2, 07100 Sassari, Italy; sgarroni@uniss.it; 4Pavia H2 Lab, C.S.G.I. & Department of Chemistry, Physical Chemistry Section, University of Pavia, 27100 Pavia, Italy; chiara.milanese@unipv.it; 5Department of Physical Chemistry of Materials, Consejo Nacional de Investigaciones Científicas y Técnicas (CONICET) and Centro Atómico Bariloche, Av. Bustillo km 9500, S.C. de Bariloche S4140, Argentina; 6Institute of Physical Chemistry, University of Hamburg, Martin-Luther-King-Platz 6, 20146 Hamburg, Germany; 7Helmut Schmidt University, University of the Federal Armed Forces Hamburg, D-22043 Hamburg, Germany

**Keywords:** confinement, confined complex hydrides, carbon-based polymer, hydrogen storage

## Abstract

In this work, the possibility of creating a polymer-based adaptive scaffold for improving the hydrogen storage properties of the system 2LiH+MgB_2_+7.5(3TiCl_3_·AlCl_3_) was studied. Because of its chemical stability toward the hydrogen storage material, poly(4-methyl-1-pentene) or in-short TPX^TM^ was chosen as the candidate for the scaffolding structure. The composite system was obtained after ball milling of 2LiH+MgB_2_+7.5(3TiCl_3_·AlCl_3_) and a solution of TPX^TM^ in cyclohexane. The investigations carried out over the span of ten hydrogenation/de-hydrogenation cycles indicate that the material containing TPX^TM^ possesses a higher degree of hydrogen storage stability.

## 1. Introduction

Hydrogen technology is considered as a key technology for ensuring the synergic exploitation of intermittent and unevenly distributed renewable energy sources. Among the features that make the use of hydrogen as energy vector extremely appealing, the most relevant is its energy density per unit of mass, which is about three times higher than that of gasoline (i.e., 141.86 MJ/kg versus 46.4 MJ/kg). However, it must be mentioned that its energy density per unit of volume is lower than that of gasoline (i.e., 10.044 MJ/L versus 34.2 MJ/L) [1,2,3,4]. In the past decades, the scientific community has been studying several possible pathways to store hydrogen in a safe as well as efficient manner [5]. In this regard, solid-state hydrogen storage, in metal hydride-based systems, presents advantages in comparison with storage systems based on compressed gaseous hydrogen and/or on liquefied hydrogen at cryogenic temperature [6,7,8]. Among the several classes of solid-state hydrogen storage materials, complex metal hydrides (e.g., borohydrides [BH_4_]^−^, alanates [AlH_4_]^−^, and amides [NH_2_]^−^) have been intensively investigated because of their high gravimetric hydrogen densities (ρ_m_), [3,4,9,10,11,12,13,14,15,16,17,18,19,20]. In fact, most of the complex hydrides display capacity values above 7 wt.% H_2_, e.g., LiBH_4_: 18.36 wt. % H_2_; LiAlH_4_: 10.54 wt. % H_2_ or LiNH_2_: 8.78 wt.% H_2_ [17]. Unfortunately, their decomposition products are too stable (B, Al, and N_2_) to re-form the starting hydrides under mild hydrogen pressure and temperatures, thus hindering their full reversibility and possibility to use them in practical applications. In addition, hydrogen is strongly bonded in anionic complexes such as [BH_4_]^−^, [NH_2_]^−^, and [AlH_4_]^−^, hence, a relatively large amount of energy is required for breaking such complexes. These energy requirements thus result in high operating temperatures and sluggish hydrogenation-dehydrogenation kinetics. Many attempts have been made to improve the hydrogen sorption properties of the complex hydride materials. The reactive hydride composite (RHC) approach (e.g., MgH_2_ + 2NaBH_4_, MgH_2_ + 2LiBH_4_ and Mg_2_NiH_4_ + Ca(BH_4_)_2_) is a method to obtain a hydrogen storage system, which still possesses a high hydrogen storage capacity, but at the same time improved thermodynamic and kinetic features (compared to the starting reactants). For example, Vajo et al. [21] destabilized LiBH_4_ by using MgH_2_, the composite LiBH_4_-MgH_2_ in the presence of 2–3 mol% TiCl_3_ reduced the overall reaction enthalpy by 25 kJ/mol of H_2_, while still having high reversible hydrogen storage capacity (8 – 10 wt.%). Similar approaches on RHC systems can be found in other studies [22,23,24,25,26]. In the last ten years, the system 2LiH+MgB_2_ has been one of the most studied RHC systems for hydrogen storage, owing to its high gravimetric hydrogen storage capacity and lower predicted thermodynamic stability (i.e., 11.5 wt.% and 45.9 kJ/mol H_2_, respectively) [25,27,28,29]. However, because of the kinetic constraints, the hydrogenation/dehydrogenation processes occur only at temperatures higher than those predicted and the reaction rate is quite slow [30,31].

A method broadly utilized to improve the kinetic behavior of complex metal hydride-based systems is the addition of appropriate transition metal (TM) and TM-based compounds, i.e., Ti-containing additives [31,32,33,34,35], Zr-containing additives [36], V-containing additives [36], Fe-containing additives [37], and Nb-based additives [33,38]. Such addition of selected additives provides a kinetically favored reaction path for the formation of reaction products, which enhance the hydrogen uptake and release. The use of a nanostructured inert scaffolding system as a tool to improve the kinetic and (in limited cases) the thermodynamic properties of complex metal hydride systems have been reported in the literature [39,40,41,42,43,44,45,46,47,48,49,50,51,52]. Several attempts have been made for the Li-RHC system, for example, Nielsen et al. [51] have used resorcinol formaldehyde aerogels (RF) with a controlled pore size of ~21 nm to embed Li-RHC, the confined Li-RHC showed a better hydrogen storage performance in terms of improved dehydrogenation kinetics and high degree of cyclic reversibility and stability versus the not Li-RHC sample, i.e., 74% of hydrogen content released at T ≤ 320 °C after 15 h; whereas the bulk Li-RHC released only 26%; and almost 92% of initial hydrogen content was maintained over three cycles under relatively milder conditions. The enhanced hydrogen storage properties for the nanoconfinement of Li-RHC are also found in the works of Gosalawit-Utke et al. [46,47,48,49,50,53], i.e., the nanoconfined Li-RHC+0.13TiCl_4_ released completely 99% of theoretical hydrogen capacity versus 94% observed for the nanoconfined Li-RHC under the same temperature: RT-500 °C and same time, 5 h at constant temperature, indicating a significant improvement in dehydrogenation kinetics. Recently, the possibility to modify the performances of amide-hydride composites via encapsulation into a polymer matrix (i.e., poly(4-methyl-1-pentene)) was also demonstrated [54]. Motivated by these positive results, it was decided to study the hydrogen sorption properties of the RHC system 2LiH+MgB_2_+7.5(3TiCl_3_·AlCl_3_) (denoted as Li-RHC_d_+7.5 (3TiCl_3_·AlCl_3_)) embedded into an inert polymer adaptive scaffold, i.e., poly(4-methyl-1-pentene), a polyolefin polymer commercially distributed by the company Mitsui Chemicals with the name TPX^TM^. The choice of this particular metal hydride composition is driven by the fact that it represents the RHC based on 2LiH+MgB_2_ with the best kinetic performances reported to date, because of the specific catalytic content in the formulation [31]. Moreover, in previous works, it was noticed that upon cycling the system 2LiH+MgB_2_ suffers a sensible and continuous decrement of the hydrogen storage capacity, because of the segregation of the two main reactants [30,55,56]. Therefore, the use of a scaffolding material can help to improve the cyclability of this system by maintaining the active components into place and confined [57], as also investigated in previously mentioned works [46,47,48,49,50]. In this work, the kinetic effects that the presence of the scaffolding structure entails on the Li-RHC_d_+7.5 (3TiCl_3_·AlCl_3_) system are investigated. The hydrogenation-dehydrogenation properties of the confined system are studied over several cycles and compared with those of not confined material. The acquired information is evaluated and discussed crosslinking the volumetric data with those obtained via morphological, microstructural, and chemical characterizations.

## 2. Materials and Methods

The reactants utilized in this work, lithium hydride (LiH, 95%, Sigma Aldrich, Darmstadt, Germany), magnesium boride (MgB_2_, 99%, Alfa Aesar, Kandel, Germany), aluminum (III) chloride-titanium (III) chloride (3TiCl_3_·AlCl_3_, with 76–78% TiCl_3_, Fisher Scientific, Schwerte, Germany) were purchased in powder form and used without further treatment.

The preparation of the Li-RHC_d_+7.5 (3TiCl_3_·AlCl_3_) followed the same procedure as described in a previous study [31]. TPX^TM^ RT18 was provided in the form of pellets from the company Mitsui Chemicals Europe Inc. GmbH, Düsseldorf, Germany. A solution of TPX^TM^ in cyclohexane was prepared under an inert atmosphere at 75 °C. The concentration of the polymer solution was 10 wt. %. The mixture of 1g Li-RHC_d_+7.5(3TiCl_3_·AlCl_3_) and 1g TPX^TM^/cyclohexane solution was milled in an 8000M Mixer/Mill^®^ High-Energy Ball Mill (SPEX SamplePrep LLC, Metuchen, NJ, USA) for 2 min in a stainless steel vial and using stainless steel balls with a ball-to-powder ratio of 10:1. Then, the materials were left to dry under argon atmosphere for 24 h. Material handling and milling were carried out in MBraun Unilab (MBraun Inertgas-system GmBH, Garching, Germany) as well as GP Campus Jacomex gloveboxes (Jacomex SAS, Dagneux, France) under a continuously purified Ar flow (O_2_ and H_2_O levels < 1 ppm).

Volumetric analyses of studied materials were carried out using a custom-made volumetric Sieverts type apparatus. The hydrogenation and dehydrogenation cycles of each sample were performed at 350 °C and 50 bar of hydrogen and 400 °C and 3 bar of hydrogen, respectively. All measurements were carried out until the measured values reached a plateau region that met the criteria of Δwt. (%) = 0.0002 and Δt (min) = 1. These data were utilized to evaluate the rate-limiting step mechanisms of sorption processes by fitting the kinetic curves with gas-solid models expressed in the reduced-time-scale form in Table 1 [58,59,60]. The best-described model provides a linear fitting with R^2^ close to 1 and a straight line through the origin with a slope of about 1. This method is known as reduced time method [61,62]. Then, if a good fit of experimental data for a specific kinetic equation is reached, the rate-limiting step of the kinetics can be determined.

XRD characterization was carried out using a Bruker D8 Discover diffractometer (Bruker AXS GmbH, Karlsruhe, Germany). This diffractometer is equipped with a copper X-ray source (Cu Kα radiation, λ = 1.54184 Å) and a VANTEC-500 area detector from Bruker. The diffraction patterns were acquired in nine steps in the 2θ range from 10° to 90° with an exposure time of 300 s per step. The preparation of the specimens to be analyzed via XRD was carried out in an MBraun Unilab glovebox under a continuously purified argon flow (O_2_ and H_2_O levels < 10 ppm). A small amount of powder (≈5 mg) was placed onto a sample holder and sealed with an airtight transparent dome made of poly-methyl-methacrylate (PMMA) in order to avoid oxidation or hydrolysis of samples during the pattern acquisition.

The composition of synthesized and cycled samples was characterized by the FT-IR technique (Cary 630 FTIR spectrometer, Agilent Technologies Deutschland GmbH, Waldbronn, Germany). The background was calibrated for each measurement. For each measurement, a small amount of material was placed on the diamond ATR top-plate, and the FT-IR spectrum was acquired in a full frequency range of 4000–400 cm^−1^ with a spectral resolution of 4 cm^−1^ and number of scans of 200.

The scanning electron microscope Merlin (Carl ZEISS, Oberkochen, Germany) was used to characterize the structure of the RHC - polymer composite material. Secondary electron images were taken at accelerating voltages. In order to avoid oxidation of the composite, the samples were prepared in a GP Campus Jacomex glovebox under Ar atmosphere (O_2_ and H_2_O levels <1 ppm), and they were transferred to the SEM with the use of vacuum transfer box (Kammrath & Weiss, Dortmund, Germany).

Differential scanning calorimetry (DSC) was performed on a DSC 1 (Star system, Mettler-Toledo, Gießen, Germany) in the temperature range of 0 °C to 300 °C, with a heating and cooling rate of 10 K/min, using nitrogen as a purge gas stream (60 mL/min). Two heating-cooling cycles were conducted per sample. For the interpretation of the results, the second heating and the cooling traces were used.

Thermogravimetric analysis (TGA) were carried out on the TG 209 F1 Iris (Netzsch, Selb, Germany). The experiments were done at a temperature range from 25 °C up to 1000 °C at a heating rate of 10 K/min. The measurements were performed under argon atmosphere.

## 3. Results

Volumetric analyses were carried out to investigate the hydrogen sorption properties and reversibility of the confined Li-RHC_d_+7.5(3TiCl_3_·AlCl_3_) over ten hydrogenation-dehydrogenation cycles. As shown in Figure 1a, the confined Li-RHC_d_+7.5(3TiCl_3_·AlCl_3_) absorbs about 7.5 wt. % H_2_ within 12 h in the 1st hydrogenation process. In the following hydrogenations, the hydrogen uptake occurs faster, i.e., 7.2 wt.% H_2_ (~95% of the hydrogen content measured in the 1st absorption) is absorbed in 1 h during the 2nd hydrogenation, and the overall hydrogenation time shrinks to about 30 min in the following cycles. During the 1st dehydrogenation of confined Li-RHC_d_+7.5(3TiCl_3_·AlCl_3_), 7.3 wt.% H_2_ (approx. 97% of the hydrogen absorbed in the hydrogenation process) is released in 45 min (Figure 2a). This value remains almost unchanged for the following dehydrogenations, i.e., the amount of desorbed hydrogen and dehydrogenation time remains nearly constant (Figure 2a). For the applied experimental conditions, these results suggest a high degree of reversibility and stability of the hydrogen storage capacity of the confined Li-RHC_d_+7.5(3TiCl_3_·AlCl_3_). For comparison, the hydrogenation and dehydrogenation properties of non-confined Li-RHC_d_+7.5(3TiCl_3_·AlCl_3_) were measured under the same experimental conditions, and are presented in Figure 1b and Figure 2b, respectively. The 1st hydrogen uptake is about 9.73 wt. % H_2_ achieved after 14 h of measurement. In the following cycles, the absorption time is shorter; that is, the hydrogenation time required to reach 95% of the maximum hydrogen content is about 30 min. This value is also necessary for dehydrogenation processes. However, the hydrogen storage capacity over the cycling period decreases gradually (Figure 2b). The significant difference in hydrogen storage capacity between the two investigated systems is due to the additional weight of the polymer in the confined Li-RHC_d_+7.5(3TiCl_3_·AlCl_3_).

To better compare and visualize the hydrogen cycling stability of these two systems, the dehydrogenation capacity versus the number of cycles was plotted in Figure 3. As it can be seen after the first cycle, the reversible hydrogen capacity of the confined Li-RHC_d_+7.5(3TiCl_3_·AlCl_3_) decreases about 0.1 wt.% H_2_ for the second dehydrogenation, and then, in the following dehydrogenation runs, it stabilizes at a value of ~7.3 wt. % H_2_. The variation of the reversible hydrogen storage capacity is about −0.005 wt. % per cycle. Consequently, the variation of hydrogen storage capacity is −0.05 wt. % over the ten cycles. For the non-confined material, the measured hydrogen storage capacity, upon cycling, first increases, reaching a maximum value of 9.5 wt. % H_2_ in the third cycle, and then starts to decrease linearly in the following cycles. After ten cycles, the loss of storage capacity is equal to 0.32 wt. % H_2_. These results suggest that the presence of the TPX^TM^ scaffold, encapsulating the Li-RHC_d_+7.5(3TiCl_3_·AlCl_3_), stabilizes the reversible hydrogen capacity without affecting the kinetics properties of the storage material.

For characterization, SEM images and XRD patterns of the non-confined and confined Li-RHC_d_+7.5(3TiCl_3_·AlCl_3_) samples after milling and after cycling were acquired. In Figure 4 and Figure 5, the corresponding SEM images and XRD patterns are shown, respectively. In Figure 4a,c, the as-milled RHC_d_+7.5(3TiCl_3_·AlCl_3_) and the as milled confined RHC_d_+7.5(3TiCl_3_·AlCl_3_) are presented, respectively. The as-milled RHC_d_+7.5(3TiCl_3_·AlCl_3_) appears to be composed mostly of particles smaller than 1 μm, which cluster to form aggregates of several tenths of μm. The as-milled confined Li-RHC_d_+7.5(3TiCl_3_·AlCl_3_) shows a morphology and particle size distribution of the RHC system similar to that of as-milled non-confined Li-RHC_d_+7.5(3TiCl_3_·AlCl_3_). Differently from the non-confined Li-RHC_d_+7.5(3TiCl_3_·AlCl_3_), the presence of the polymeric matrix enveloping the RHC system is also visible (Figure 4c,d). It must be mentioned that the TPX^TM^ polymer does not cover the RHC particles completely, but rather forms an open matrix structure, which partially envelops the RHC particles, validating the hypothesis of the polymer scaffold formation.

Nevertheless, this scaffold is not forming a continuous phase but holds the RHC particles in a confined space. The morphology of the cycled non-confined and confined samples appears different from the one of the as-milled, respectively, in Figure 4b,e. The particles smaller than 1 μm are still present, but in a smaller number, and the polymeric matrix of the confined material is not discernible from the RHC material. To better visualize the morphological differences between the two cycled samples, a smaller magnification of the SEM pictures was chosen. The SEM pictures of the cycled non-confined and of the cycled confined material (Figure 4b,e) show the coexistence of two utterly different material morphologies. A portion of the material still appears to be composed of large aggregates (of the overall dimension of several tenths of μm) of smaller particles. However, for both cycled samples, large portions of material appear to be composed of a solidified molten phase. In order to investigate the status of the non-continuous polymer scaffold after cycling, a higher magnification SEM picture was taken (Figure 5a). In order to help to localize carbon-based compounds, an EDX elemental mapping of the same area was also taken (Figure 5b, EDX spectrum- ESI Appendix A). As in the case of the SEM pictures collected at smaller magnification for both cycled materials, two distinct material portions can be distinguished. Interestingly, the molten-like fraction of the sample appears to contain carbon, whereas the grainy fraction appears to contain mostly Mg. Unfortunately, because of the low sensitivity of the EDX technique to light elements, such as Li and B, it is not possible to map their distribution in the sample. Therefore, it is possible to assume that the grainy fraction contains, indeed, all the elements of the Li-RHC, even if only Mg is detected.

Figure 6 shows the XRD patterns of the nonconfined and confined Li-RHC_d_+7.5(3TiCl_3_·AlCl_3_) samples. In Figure 6A(a), the diffraction pattern of the non-confined Li-RHC_d_+7.5(3TiCl_3_·AlCl_3_) after milling shows the presence of MgB_2_ (~73 wt. %), LiH (~26 wt. %), and LiCl (~1 wt. %). The diffraction peaks of these phases are also found in the sample after cycling (Figure 6 A(b)), the side product LiCl accounts for about 14.4 wt.% based on the Rietveld refinement. The formation of LiCl reduces the theoretical hydrogen storage capacity of the whole system. Noticeably, the intensity of LiCl diffraction peaks increased, suggesting a further reaction between Cl^−^ and Li^+^ during cycling or an increment of the average crystallite size of LiCl. A similar observation is found for the confined Li-RHC_d_+7.5(3TiCl_3_·AlCl_3_) material (Figure 6B), the diffraction intensity of LiCl increases over cycling, approx. 12.8 wt.% of LiCl is detected in the cycled confined material (Figure 6B(b)). No difference in XRD patterns of the studied materials (with and without TPX^TM^, after milling and after cycling) indicates that the presence of TPX^TM^ polymer does not affect the reaction path of Li-RHC_d_+7.5(3TiCl_3_·AlCl_3_) by forming no crystalline compounds.

To investigate the stability and role of the TPX^TM^ scaffold on the enhancement of the hydrogen storage performance of the system Li-RHC_d_+7.5(3TiCl_3_·AlCl_3_), the material before and after cycling was investigated using FT-IR spectroscopy (Figure 7A,B). The FT-IR spectrum measured for 3TiCl_3_·AlCl_3_ is presented for comparison, as well. For the samples after milling (Figure 7A(a,b)), an absorption band centered at 1132 cm^−1^ can be attributed to the B-H bending vibration or the B–O stretching of tetrahedral BO_4_^-^ structural units. In the cycled materials (Figure 7A(c,d)), in both systems, the band at 1132 cm^−1^ is not visible anymore. In the as-prepared confined Li-RHC_d_+7.5(3TiCl_3_·AlCl_3_) sample (Figure 7A(b),B(b)), a small signal at around 2960 cm^−1^ is visible. This signal is most likely due to the presence of TPX^TM^ [63]. Interestingly, after cycling (Figure 7A(d)), this peak is not visible anymore. This result suggests that the TPX^TM^ degrades during cycling. However, because of the weak nature of the related FT-IR signals, the determination of the degradation products is not possible. Although in the cycled samples the presence of Li_2_B_12_H_12_ is likely, the FT-IR absorption band of [B_12_H_12_]^−2^ at around 2480 cm^−1^ is not observed. This result could be due to an overlap with the broad absorption band generated by the ATR diamond window of the instrument.

In order to characterize the chemical evolution of B-containing species in the system containing TPX^TM^, several attempts to perform solid-state magic-angle-spinning NMR were made, but without any success. The pure solubility of TPX at room temperature in common organic solvents was an additional factor that caused this examination not possible.

## 4. Discussion

The volumetric measurements performed in this work hinted to an improvement of the hydrogen storage stability of the system Li-RHC_d_+7.5(3TiCl_3_·AlCl_3_) by the embedment of the same into a TPX^TM^ matrix. The decrement of the hydrogen storage capacity after the 10th cycle for the TPX^TM^ confined Li-RHC_d_+7.5(3TiCl_3_·AlCl_3_) is 0.05 wt. % whereas for the not confined system is 0.32 wt. %. Even considering the higher starting hydrogen capacity of the pristine system, the embedded sample ensures a significantly longer lifespan of the material, indicated as thousands of cycles by targets and requirements [64]. Moreover, the stability of the reversible hydrogen of Li-RHC_d_+7.5(3TiCl_3_·AlCl_3_) confined in the TPX^TM^ is slightly better in comparison to the other confined systems, for example, more than 95% initial hydrogen storage capacity is preserved over ten dehydrogenation cycles, which is significantly higher than that of the LiBH_4_ confined in 25 nm aerogel (~40% through three cycles) [41] and of the composite LiBH_4_+MgH_2_ confined in resorcinol formaldehyde aerogels (~92% through three cycles) [51]. The reason behind this phenomenon can be explained by the presence of a TPX^TM^ surrounding the hydrogen storage system. At the beginning of this manuscript, it was mentioned that the adaptation of the RHC system to the polymer material has led to the formation of an adaptive non-continuous scaffold. Differently from the traditional scaffolding material that has a predetermined rigid structure, the use of a polymer solution leads to the adaptation of the storage material independently from their particle size and particle shape. Thus, all the particles of the Li-RHC_d_+7.5(3TiCl_3_·AlCl_3_) benefit from the presence of the scaffold, which limits potential phase separations as well as avoids agglomeration phenomena. As a result of the presence of the scaffolding material, close contact between LiH and MgB_2_ can be maintained upon cycling. In this regard, the SEM observation reported in Figure 4 supports the initial assumptions. The TPX^TM^ polymer forms a matrix that surrounds non-continuously the RHC material. This matrix upon cycling changes (Figure 4d,e and Figure 5b), most likely because of the decomposition of TPX^TM^ at high temperatures (TPX^TM^ [54] is stable up to approx. 350 °C, while it melts at around 230 °C, ESI Appendix A). However, the carbonaceous product of TPX^TM^ decomposition appears to maintain the non-continuous scaffold structure that plays a key role in maintaining the hydrogen storage properties of the RHC material.

The decomposition of the TPX^TM^ during cycling is also suggested by the FT-IR measurements (Figure 7). The FT-IR spectrum collected for the confined Li-RHC_d_+7.5(3TiCl_3_·AlCl_3_) after milling presents a clear absorption band at around 2960 cm^−1^, which can be related to the asymmetric bonds of CH_3_ [63]. After cycling, this signal is no longer visible. For the milled samples (Figure 7A(a,b)), FT-IR shows an absorption band at 1132 cm^−1^. This band is most likely due to the B–O stretching vibration of tetrahedral BO_3_^−^/BO_4_^−^ groups stemming from impurities present in the as-received MgB_2_ (e.g., B_2_O_3_). Upon cycling, these impurities disappear (Figure 7(c,d)). This outcome is most likely due to the reaction, at high temperature, of the BO_3_^−^/BO_4_^−^ groups with MgH_2_ to form nanocrystalline MgO.

The chemical stability of TPX^TM^ and its decomposition products toward the RHC system was also investigated. According to XRD results (Figure 6), MgB_2_, LiH, and LiCl are present in both the non-confined and confined Li-RHC_d_+7.5(3TiCl_3_·AlCl_3_) materials after milling and after the 10th cycle. The XRD diffractograms of these two materials are alike in both milled and cycled samples. This result suggests that the scaffold does not affect in any way the reaction pathway [19].

In order to examine whether the presence of TPX^TM^ might influence the rate-limiting step mechanism of hydrogenation and dehydrogenation of Li-RHC_d_+7.5(3TiCl_3_·AlCl_3_), the Sharp and Jone’s method for modeling gas-solid reaction kinetics (Table 1) has been applied [61,62]. In this method, the experimental values ((t/t_0.5_)_experimental_) versus the theoretical ones ((t/t_0.5_)_theoretical_) are plotted, it assumes that the most appropriate reaction model meets essential criteria of a linear fitting with R^2^ close to 1 and a slope of about 1; then, if a good fit of experimental data concerning a specific kinetic equation is obtained, the rate-limiting step of kinetics can be determined. According to the rate equations of gas-solid models given in Table 1, different equations are applied to the experimental data of the second hydrogenation for the Li-RHC_d_+7.5(3TiCl_3_·AlCl_3_) and Li-RHC_d_+7.5(3TiCl_3_·AlCl_3_) with TPX^TM^ samples. As examples of the fitting process (Figure 8—Table 2, Figure 9—Table 3), the hydrogenation behavior of both Li-RHC_d_+7.5(3TiCl_3_·AlCl_3_) and Li-RHC_d_+7.5(3TiCl_3_·AlCl_3_) plus TPX^TM^ samples are best described by the three-dimensional interface-controlled reaction (R3). Although the model R2 and R3 (in Table 2), as well as F1 and R3 (in Table 3), do not present extensive differences in the fitting criteria, R3 model was chosen as the most suitable one for this hydride system, in both instances. A typical example of this model can be found in the work of Bösenberg et al. [32]; the hydrogenation behavior of LiH-MgB_2_ with various transition metal additives (V, Ti) fits well the three-dimensional interface controlled reaction; the interface reaction during decomposition of MgB_2_ is found to be the rate-limiting step and the microstructures of MgB_2_ plays a crucial role in the phase transformation process. In the present work, the same approach was used to determine the mechanism of hydrogenation process and the obtained results are in good agreement with the previously published works [32,37]. Hence, the possibility to fit both the experiments with the model R3 indicates that the presence of the polymer scaffold does not influence the reaction mechanism of the material. Therefore, the beneficial effect on the hydrogen storage capacity observed upon cycling can be attributed only to the confinement of the system, which hinders the large-scale segregation of system components.

## 5. Conclusions

The mixing via ball-milling of Li-RHC_d_+7.5(3TiCl_3_·AlCl_3_) powder and a solution of TPX^TM^ in cyclohexane resulted in the formation of a solid composite, where the RHC particles are embedded into a polymeric matrix, which acts as a non-continuous polymer scaffold. In comparison to the non-confined material, the confined material exhibits a higher degree of hydrogen storage stability upon cycling. The decrement of the hydrogen storage capacity after the 10th cycle for the confined Li-RHC_d_+7.5(3TiCl_3_·AlCl_3_) is about six times lower than that of the non-confined system. This result might be attributed to the presence of the scaffolding structure surrounding the RHC system that hinders the long-range phase segregation. It was observed that because of the relatively high operative temperature of the system Li-RHC, the TPX^TM^ decomposes, but maintains scaffolding properties along with the positive effect on the hydrogen storage stable cycling of the composite. The XRD analyses and the result of the kinetic modeling indicate that the presence of the polymeric scaffold does not modify the reaction mechanism and/or influence the system rate-limiting steps.

## Figures and Tables

**Figure 1 materials-13-00991-f001:**
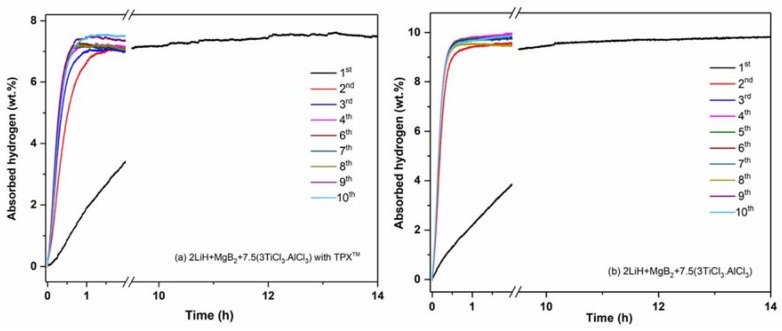
Hydrogenation profiles and cycling capability of the composites Li-RHC_d_+7.5(3TiCl_3_·AlCl_3_) with TPX^TM^ (**a**) and without TPX^TM^ scaffold (**b**).

**Figure 2 materials-13-00991-f002:**
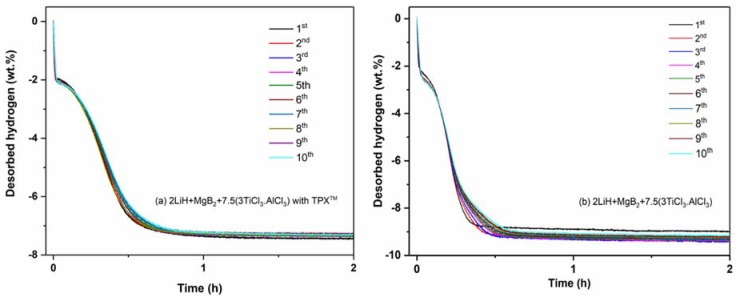
Dehydrogenation profiles and cycling capability of the composites Li-RHC_d_+7.5(3TiCl_3_·AlCl_3_) with TPX^TM^ (**a**) and without TPX^TM^ scaffold (**b**).

**Figure 3 materials-13-00991-f003:**
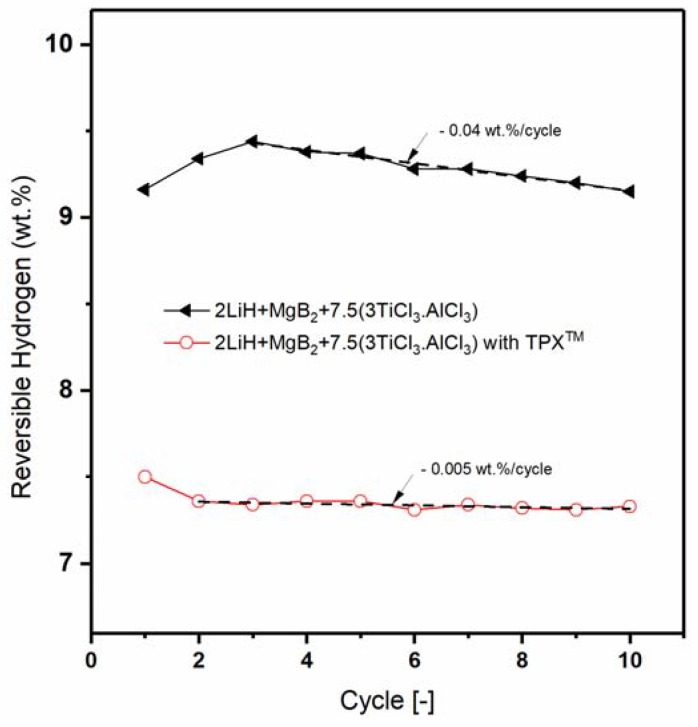
Reversible hydrogen capacity for dehydrogenation over the cycling of the confined and non-confined Li-RHC_d_+7.5(3TiCl_3_·AlCl_3_) materials.

**Figure 4 materials-13-00991-f004:**
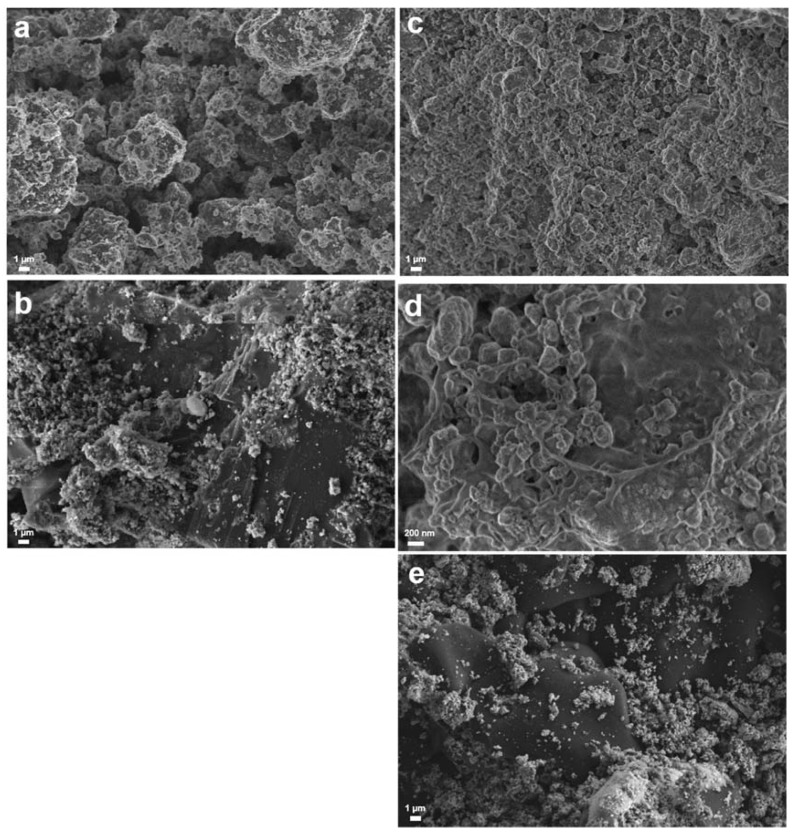
SEM images of non-confined Li-RHC_d_+7.5(3TiCl_3_·AlCl_3_): (**a**) as-milled and (**b**) as-cycled material; and Li-RHC_d_+7.5(3TiCl_3_·AlCl_3_)+TPX^TM^; (**c**,**d**) as-milled material at different magnifications, (**e**) as-cycled material.

**Figure 5 materials-13-00991-f005:**
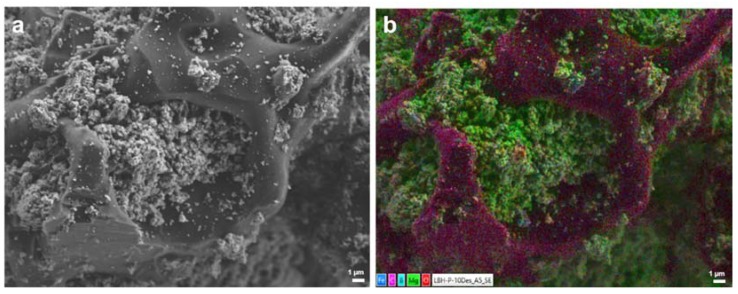
SEM image (**a**) and EDX elemental mapping (**b**) of the cycled RHC_d_+7.5(3TiCl_3_·AlCl_3_)+TPX^TM^ (Fe, C, B, Mg, and O).

**Figure 6 materials-13-00991-f006:**
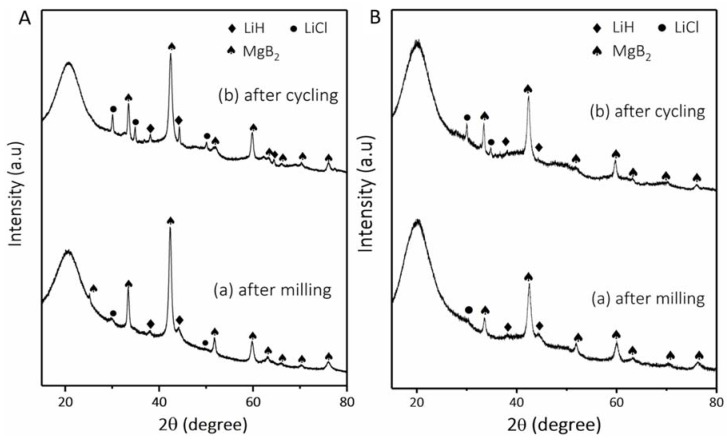
XRD patterns of the composites (**A**) Li-RHC_d_+7.5(3TiCl_3_·AlCl_3_): (a) after milling and (b) after cycling; and (**B**) Li-RHC_d_+7.5(3TiCl_3_·AlCl_3_) with TPX^TM^: (a) after milling and (b) after cycling.

**Figure 7 materials-13-00991-f007:**
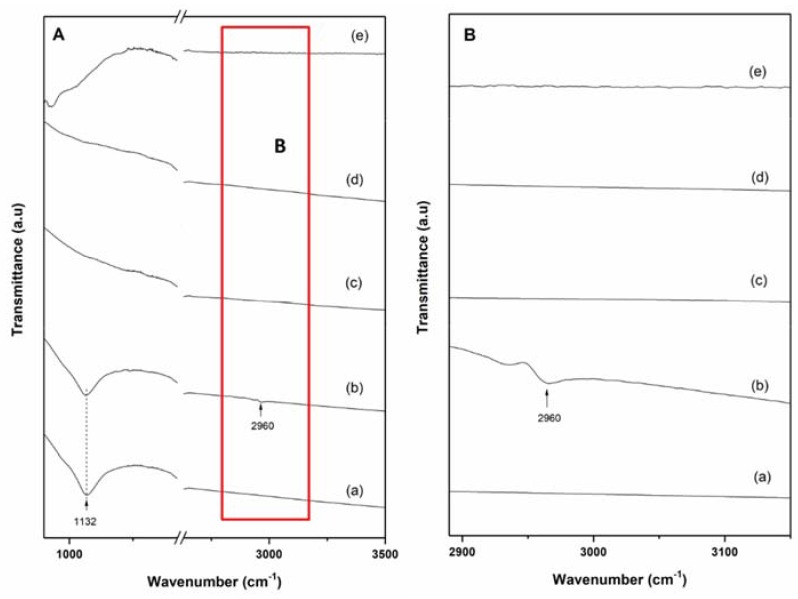
FT-IR spectrum of the non-confined and confined Li-RHC_d_+7.5(3TiCl_3_·AlCl_3_) materials: (**A**) (a) Li-RHC_d_+7.5(3TiCl_3_·AlCl_3_) after milling; (b) Li-RHC_d_+7.5(3TiCl_3_·AlCl_3_) with TPX^TM^ after milling; (c) Li-RHC_d_+7.5(3TiCl_3_·AlCl_3_) after 10^th^ cycle; (d) Li-RHC_d_+7.5(3TiCl_3_·AlCl_3_) with TPX^TM^ after 10^th^ cycle; and (e) pure 3TiCl_3_·AlCl_3_. (**B**) Zoom of (**A**) between 2890 to 3150 ppm.

**Figure 8 materials-13-00991-f008:**
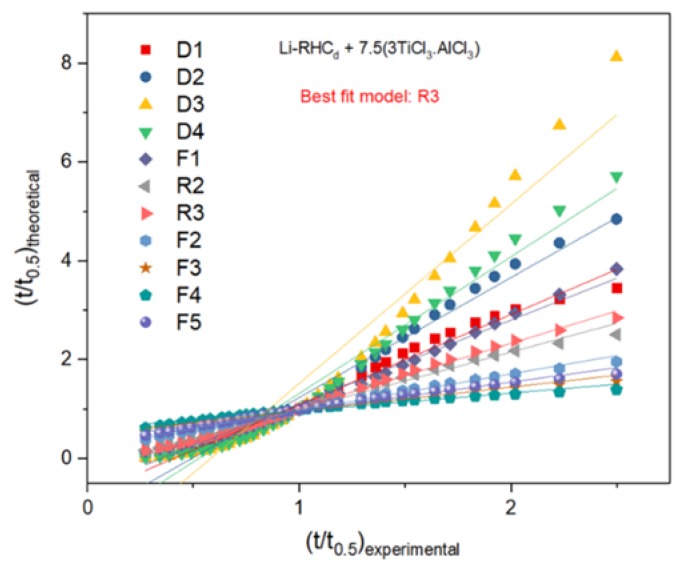
(t/t_0.5_)_experimental_ vs. (t/t_0.5_)_theoretical_ plots for Li-RHC_d_+7.5(3TiCl_3_·AlCl_3_) at the 2nd hydrogenation at 400 °C and under 50 bar of H_2_. The method assumes that the most suitable reaction model provides a linear fitting with R^2^ close to 1 and a slope of about 1.

**Figure 9 materials-13-00991-f009:**
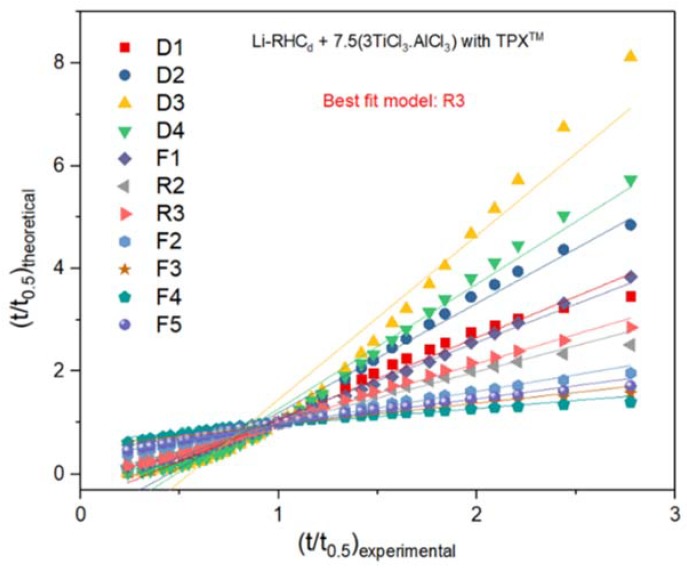
(t/t_0.5_)_experimental_ vs. (t/t_0.5_)_theoretical_ plots for Li-RHC_d_+7.5(3TiCl_3_·AlCl_3_) with TPX^TM^ at the 2nd hydrogenation at 400 °C and under 50 bar of H_2_. The method assumes that the most suitable reaction model provides a linear fitting with R^2^ close to 1 and a slope of about 1.

**Table 1 materials-13-00991-t001:** Summary of integral expressions for different reaction models [59,60,61,62].

Kinetic Rate Models	Rate Equations in the Reduced-Time-Scale Form
D1 one-dimensional diffusion	α^2^/0.25
D2 two-dimensional diffusion	[α + ((1 − α)ln(1 − α))]/0.1534
D3 Jander eq. for three-dimensional diffusion	[(1 − (1 − α)^1/3^)^2^]/0.04255
D4 Ginstling-Braunshtein eq. for three-dimensional diffusion	[1 − 2/3α − (1 − α)^2/3^]/0.0367
R2 two-dimensional interface controlled	[1 − (1 − α)^1/2^]/0.29289
R3 three-dimensional interface controlled	[1 − (1 − α)^1/3^]/0.20629
F1 JMA − n = 1	−ln(1 − α)/0.6931
F2 JMA − n = 1/2	−ln(1 − α)^1/2^/0.832
F3 JMA − n = 1/3	−ln(1 − α)^1/3^/0.8849
F4 JMA − n = 1/4	−ln(1 − α)^1/4^/0.9124
F5 JMA − n = 2/5	−ln(1 − α)^2/5^/0.8636

**Table 2 materials-13-00991-t002:** Fitting parameters extracted from corresponding applied models (referred to Figure 8).

Fraction from 0.10 to 0.93	Intercept Value	Intercept Error	Slope Value	Slope Error	Statistics Adj. R-Square
D1 one-dimensional diffusion	−0.73706	0.05423	1.82867	0,04359	0.98213
D2 two-dimensional diffusion	−1.18716	0.08155	2.42902	0.06556	0.97721
D3 Jander eq. for three dimensional diffusion	−2.10247	0.2053	3.6289	0.16504	0.9378
D4 Ginstling-Braunshtein eq. for three dimensional diffusion	−1.44381	0.10988	2.76764	0.08833	0.9684
F1 JMA − n = 1	−0.58708	0.04133	1.69726	0.03323	0.98788
R2 two-dimensional phase	−0.17235	0.02398	1.16644	0.01928	0.99133
R3 three-dimensional phase boundary	−0.28582	0.001866	1.11148	0.0015	0.99583
F2 JMA − n = 1/2	0.23639	0.01434	0.74068	0.01153	0.99231
F3 JMA − n = 1/3	0.48376	0.01625	0.48267	0.01306	0.97709
F4 JMA − n = 1/4	0.60815	0.01487	0.3597	0.01195	0.96584
F5 JMA − n = 2/5	0.38489	0.01619	0.58345	0.01301	0.98432

**Table 3 materials-13-00991-t003:** Fitting parameters extracted from corresponding applied models (referred to Figure 9).

Fraction from 0.10 to 0.93	Intercept Value	Intercept Error	Slope Value	Slope Error	Statistics Adj. R-Square	
D1 one-dimensional diffusion	−0.5432	0.04743	1.59934	0.03605	0.984	
D2 two-dimensional diffusion	−0.93653	0.06158	2.13041	0.0468	0.98478	
D3 Jander eq. for three dimensional diffusion	−1.74044	0.16713	3.19363	0.12701	0.95175	
D4 Ginstling-Braunshtein eq. for three dimensional diffusion	−1.16156	0.08411	2.43031	0.06392	0.97833	
F1 JMA − n = 1	−0.4105	0.02734	1.48735	0.02078	0.99379	
R2 two-dimensional phase	−0.04597	0.0256	1.01779	0.01945	0.98844	
R3 three-dimensional phase boundary	−0.14561	0.01717	1.04599	0.01305	0.99587	
F2 JMA − n = 1/2	0.31754	0.01728	0.64549	0.01313	0.98692	
F3 JMA − n = 1/3	0.53768	0.01789	0.41974	0.0136	0.96747	
F4 JMA − n = 1/4	0.64873	0.01593	0.3124	0.01211	0.9541	
F5 JMA − n = 2/5	0.44955	0.01835	0.50783	0.01395	0.97641

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
