# Peer review of "Enhanced Stability of Li-RHC Embedded in an Adaptive TPX™ Polymer Scaffold"

_materials, 2020, doi:10.3390/ma13040991_

Round 1

Reviewer 1 Report

The metal-polymer nanocomposite holds great promise as a general approach for the future hydrogen storage system, as the metal-polymer is proposed to simultaneously provide air-stability, high hydrogen storage density, and rapid hydrogen kinetics.

Nanocomposites where nanomaterials are incorporated with polymers or porous carbon structures have recently piqued significant interest attributed to their superior optical, electrical, mechanical and gas-sensing properties. Extensive research efforts have demonstrated that nanostructuring, nanoconfinement and formation of nanocomposites give rise to the increased specific surface area, as well as increased concentration of defects, steps, and corner atoms. These effects result in the enhanced kinetics attributed to the shorter diffusion lengths for H2 and possibly higher nucleation rates and better reversibility and thermal stability during cycling specifically.

For example, LiBH4/MgH2 mixtures confined within nanoporous carbon aerogels showed the enhanced kinetics rates and the significantly reduced dehydrogenation temperature[1, 2]. Mg nanocrystals in a poly(methyl methacryslate) matrix (Mg-PMMA) has been highlighted as it has been proved to improve the kinetics, have high stability against moisture and O2 for extended cycles, and a high degree of reversibility[3].

This manuscript proposed a polymer-based adaptive scaffold for improving hydrogen storage properties of the Li-based reactive hydride composite. The research idea that ball-milled Li-RHC with a solution of TPXTM was novel and original. However, the Li-RHC/TRX nanocomposite system does how a significant on the hydrogen storage properties in terms of hydrogen capacity, kinetics and cyclability.

1, in the 262 line, the author claimed the hydrogen storage capacity of the Li-RHCd+7.5(3TiCl3·AlCl3) was improved by the embedment into a TPX matrix. However, the capacity of Li-RHCd+7.5(3TiCl3·AlCl3)/TRX is 7.5 wt.% and the capacity of Li-RHCd+7.5(3TiCl3·AlCl3) without TRX is 9.73wt.%.

Considering the -0.04% per cycle on H2 capacity reduction, it takes over 60 cycles for Li-RHCd+7.5(3TiCl3·AlCl3) to reach the same H2 capacity as Li-RHCd+7.5(3TiCl3·AlCl3)/TRX that has -0.005 wt.% H2 loss per cycle.

2, Figure 1 b shows the hydrogen capacity increased with hydrogenation and dehydrogenation cycles. Obviosity, the H2 capacity of the 10th cycle is higher than 3rd, 2nd and 8th cycles. This data conflicts with the dehydrogenation cycling capability that is reduced during cycles (in Figure 2b).

3. As the TRX decomposed during cycles, it can hardly tell the effects of the decomposition from TPX and the temperatures. The TGA of TRX is highly recommended to be carried out to confirm the decomposition, melting and burning temperature of the TRX.

Overall, I would not recommend the paper is accepted to be published in Materials because the hydrogen storage capacity of the original system is reduced over 20 wt.% by introducing TRXTM and It is not demonstrated the positive effect of TRXTM on hydrogenation and dehydrogenation kinetics, neither cyclabilities. Maybe, the TRX slightly improved the dehydrogenation cyclabilities; however, comparing with the significantly reduced H2 capacity, the outcome of introducing TRXTM needs to be carefully considered.

Nielsen, T.K., et al., A Reversible Nanoconfined Chemical Reaction. ACS Nano, 2010. 4(7): p. 3903-3908. Tian, M. and C. Shang, Mg-based composites for enhanced hydrogen storage performance.International Journal of Hydrogen Energy, 2019. 44(1): p. 338-344. Jeon, K.-J., et al., Air-stable magnesium nanocomposites provide rapid and high-capacity hydrogen storage without using heavy-metal catalysts. Nature Materials, 2011. 10(4): p. 286-290.

Reviewer 2 Report

The manuscript by Le et al. presents that a polymer-based adaptive scaffold, TPXTM,  was chosen as the candidate for the scaffolding structure, for improving the hydrogen storage properties of the system 2LiH+MgB2+7.5(3TiCl3·AlCl3). The composite system was obtained after ball milling of 2LiH+MgB2+7.5(3TiCl3·AlCl3) and a solution of TPXTM in cyclohexane. The investigations carried out over the span of 10 hydrogenation/de-hydrogenation cycles indicate that the material containing TPXTM possesses a higher degree of hydrogen storage stability.

The technical and scientific description in the manuscript is justified and self-consistent. The language is clear and fully understandable. However, the reviewer is not firmly convinced that the materials containing TPX possesses a higher degree of hydrogen storage stability.

The manuscript is lacking of comparisons, either from literature or experiment, with similar category of materials for hydrogen storage. The performance of hydrogen storage is mainly focused on the dehydrogenation, not the hydrogenation, which is equally important. In Figure one, the one without TPX scaffold seems to be better in terms of hydrogenation, i.e. 9.73% versus 7.2 % for non-confined versus confined. In addition, the authors claimed that the loss of capacity is 0.32wt% H2 versus 0.05% for non-confined versus confined, yes, it seems better, is very small, especially when normalized to the reversible hydrogen percent, i.e., 0.32/9.73 = 0.03288 versus 0.05/7.2=0.0069. Perhaps, literature values can be used for comparison. Kinetics data: R2 and R3 are not much of the difference in Table 2:, same as F1 JMA and R3 in table 3. The determination of kinetic model R3 needs to be justified using other methods or explained.

The manuscript does meet the topics covered by Materials.

Reviewer 3 Report

The hydrogen energy is a promising new energy technology that can develop the renewable energy with no intermittent problems and unevenly distributed sources. This manuscript “Enhanced stability of Li-RHC embedded in an adaptive TPX™ polymer scaffold” reports a polymer-based adaptive scaffold for improving the hydrogen storage properties of 2LiH+MgB2+7.5(3TiCl3·AlCl3). By ball milling 2LiH+MgB2+7.5(3TiCl3·AlCl3) with a solution of TPXTM in cyclohexane, the solid composite exhibits a higher degree of hydrogen storage stability upon cycling. However, the reference citation is a major issue that needs to be addressed before the consideration of publishing.

(1) Please check again the citation information of reference 6. What is the year of publication? Please check again the titles of the book and its chapter cited.

(2) What are the gravimetric and volumetric hydrogen storage capacities reported in reference 3, 4, and 9-20? What are the advantages and disadvantages of these complex metal hydrides? How these solid-state hydrogen storage materials build up the research track? Please give a clear summary of the cited references rather than grouping them and throwing to the readers. The wrapped references [21 - 26], references [27 - 33], references [34 - 47] have the same problems.

(3) Give suitable references to support the statement on line 57 - line 62 and line 77 - line 80. Section 4. Discussion lacks of reference papers to support the discussion made.

(4) These data in Table 1 that are utilized to evaluate the rate-limiting step mechanisms of sorption processes are collected from references 49 - 51. Why the citation marked in Table 1 is references 50 - 53?

(5) What is the EDX spectrum of Fig. 5b? Please provide the spectrum in support of the EDX elemental mapping result.

Round 2

Reviewer 3 Report

The authors have addressed the major concerns raised by the reviewer. They have provided point to point responses and have also revised the manuscript. I would suggest the consideration of the revised manuscript for publishing in Materials as an Article.